# An integrated modeling approach to evaluate the impacts of nature-based solutions of flood mitigation across a small watershed in the southeast United States

Betina I. Guido[1], Ioana Popescu[1], Vidya Samadi[2], Biswa Bhattacharya[1]

[1]Department of Hydroinformatics and Socio-Technical Innovation, IHE Delft Institute for Water Education, Delft, 2611 AX, the Netherlands.
[2]Department of Agricultural Sciences, Clemson University, Clemson, SC 29634, United States.

*Correspondence to*: Betina Guido (betinaguido@gmail.com)

**Abstract.** Floods are among the most destructive natural hazards in the world, posing numerous risks to societies and economies globally. Accurately understanding and modeling floods driven by extreme rainfall events has long been a challenging task in the domains of hydrologic science and engineering. Unusual catchment responses to flooding cause great difficulty in predicting the variability and magnitude of floods, as well as proposing solutions to manage large volumes of overland flow. The usage of Nature-Based Solutions (NBS) has proved to be effective in the mitigation of flood peak rate and volume in urban or coastal areas, yet it is still not widely implemented due to limited knowledge and testing compared to traditional engineering solutions. This research examined an integrated hydrological and hydraulic modeling system to understand the response of an at-risk watershed system to flooding and evaluate the efficacy of NBS measures. Using the Hydrologic Engineering Center Hydrologic Modeling System and River Analysis System (HEC-HMS and HEC-RAS) software, an integrated hydrologic-hydraulic model was developed for Hurricanes Matthew (2016) and Florence (2018) driven floods across the Little Pee Dee-Lumber Rivers watershed, North and South Carolina (the Carolinas), USA. The focus was on Nichols town, a small town that has been disproportionately impacted by flooding during these two hurricane events.

The present article proposes a methodology for selecting, modeling, and evaluating the performance of NBS measures within a catchment, which can be extended to other case studies. Different NBS measures including flood storage ponds, riparian reforestation, and afforestation in croplands were designed, modeled, and evaluated. Hurricane Matthew's flooding event was used for evaluating the NBS scenarios given its high simulation accuracy in flood inundation compared to the less accurate results obtained for Hurricane Florence. The scenario comparison evidenced that large-scale natural interventions, such as afforestation in croplands, can reduce the inundated area in Nichols town by 8% to 18%. On the contrary, the smaller-scale interventions such as riparian reforestation and flood storage ponds showed a negligible effect of only 1% on flood mitigation.

**Keywords**: Nature-Based Solution; Flood Risk Mitigation; Flood Hydrograph Simulation; Hurricane Matthew; Hurricane Florence; The Carolinas.

 **1 Introduction**

Floods are among the most destructive natural hazards in the world, posing numerous risks to societies and economies globally (European Parliament, 2017; IPCC AR6, 2022). The socioeconomic impacts of flooding are numerous, negatively affecting human life, health, livelihoods, and critical infrastructure, among others (Phillips et al., 2017; IPCC AR6, 2022).

In the United States, flooding and severe storms are among the most recurrent weather and climate disasters, which caused $492 billion in economic damages in the past 30 years (NOAA National Centers for Environmental Information, 2022). The US Gulf and East Coast are vulnerable to destructive tropical storms and hurricanes, which can generate storm surges and riverine flooding along the exposed communities (NOAA, 2020). Since the 1970s inland flooding has been responsible for more than half of all deaths associated with tropical cyclones in the United States. (NOAA, 2018). The state of South Carolina (SC) alone has suffered almost $7 billion in floods and hurricane damages in the last 25 years (SCDNR, 2020). This state was recently overwhelmed by the impacts of five hurricanes in a time span of only seven years, which generated severe riverine floods across cities and towns (Williams et al., 2019). According to the Intergovernmental Panel on Climate Change (IPCC) reports (e.g., IPCC AR6, 2022), there will be an increase in the tropical cyclone rainfall intensities, as well as in the proportion that reaches very intense levels (Category 4 and 5) as a result of climate change (Knutson, 2022; Stone and Cohen, 2017). In addition, it is expected that global warming would also reduce the speed of tropical cyclones, resulting in more precipitation falling over a longer period, thus increasing flooding hazards (Kossin, 2018).

On top of the climatic causes of riverine flooding, it is expected that the future development of floodplains will raise the number of houses and citizens at risk in SC (SCDNR, 2020). Many of the new constructions of the last decade in the state were built in proximity to permanent water bodies (Tedesco et. al., 2020). In SC, wetlands, and forests have experienced significant losses since historic dates (Fish and Dahl, 1999). There are several indications that the loss of these ecosystems, along with a growth in urbanization, are key drivers of increasing flood peaks and volumes in rivers and floodplains (Bronstert et al., 2002).

Systematic efforts within the research community have been made to simulate hurricane-driven floods (Chen et al., 2021; Wing et al., 2019; among others). Accurate flood prediction and inundation mapping are vital for improving forecasts, resiliency, and reducing economic damages caused by extreme events (Grimaldi et al., 2019). However, understanding and accurately predicting major floods driven by hurricanes has long been a challenge in the fields of hydrologic science and engineering (Phillips et al., 2017; NOAA, 2018; Wing et al., 2019). The spatio-temporal variability of hurricane events, rapid and unusual catchment responses to flooding, and various sources of model errors make accurate flood modeling a challenging task (Teng et al., 2017; Mohamad et al., 2021; Zhou et. al., 2020). Numerous improvements were developed for increased accuracy in flood inundation modeling and mapping over the years (Abbott et al., 1986; Dutta et al., 2006, 2013) Among others, remote sensing technologies have gained popularity in recent years, which are mainly used as an aid for flood simulation and mapping to validate and calibrate hydrological and hydraulic models (Chen et al., 2021; Teng et al., 2017).

Given the complexity of flood prediction and the increased flood hazard due to regional and global changes, an integrated physics-based flood model would overcome barriers in designing appropriate modeling architectures to represent rainfall-flooding processes. Hydrologic and hydraulic modeling integration has the benefit of using up-to-date software to model the dynamics of extreme events (Anselmo et al., 1996). This implies using a range of atmospheric, hydrologic, and hydraulic data and encompassing several processes across the water cycle. This research explores how to represent these complex processes in an integrated physics-based fashion to accurately reproduce past hurricane-driven floods.

The present research focuses on a small watershed, the Little Pee Dee (LPD) and Lumber Rivers watershed, affluents to the Pee Dee River in eastern North and South Carolina (the Carolinas). Nichols town, located at the confluence of Little Pee Dee and Lumberton Rivers, is one of the severely impacted regions in this watershed. The town was devastated by Hurricane Matthew in October 2016 and, when the residents were still recovering, Hurricane Florence ravaged the community once more in September 2018 (Edwards, 2020; Stewart and Berg, 2019). Despite the widespread devastation in the area, it is unlikely that any flood protection measures will be implemented in the short term since no advanced studies of the river's hydrodynamic behavior have been conducted so far. In light of this, there is an undeniable need for flood research that can provide insights into extreme hazards and flood risk management to protect the region from current and future damage.

Previous research has evaluated grey engineering floods mitigation measures such as levees and the elevation of bridges to protect Nichols town (Muller, 2020). Although these traditional engineering measures are widely evaluated and can significantly reduce local flood hazards, they are often categorized as an expensive and inflexible approach (Brink et al., 2016). Nature-based solutions (NBS), on the contrary, have been gaining popularity in the past years given their capacity to mitigate flood hazards and improve resilience in cost-effective ways (Cohen-Shacham et al., 2016; EESI, 2019; Ruangpan, 2019). In an area strongly affected by climatic disasters which are expected to increase due to climate change and urbanization, it is crucial to explore a more nature-based approach that can adapt to future alterations and provide benefits at a watershed level (Kalantari et al., 2018).

NBS can decrease and/or delay floodwater peaks and volumes using natural processes, reduce the magnitude of riverine floods, and increase the lead time to give more time for emergency response (Lama et al., 2021). The International Union for Conservation of Nature (IUCN), the World Bank Group, and the World Resources Institute (WRI) define NBS as "actions to protect, sustainably manage, and restore natural or modified ecosystems that address societal challenges effectively and adaptively, simultaneously providing human well-being and biodiversity benefits" (IUCN, 2022). Such solutions can improve not only flood risk but also combat climate change, improve water quality, restore, and protect wetlands, stabilize shorelines, and reduce urban heat island effects, among others (European Commission, n.d.). The use of these kinds of solutions in an area where nature is one of its main attractions such as Nichols town is imperative.

Even though the concept of NBS is well known, NBS incorporation into hydrologic and hydraulic models is not yet well understood and explored in the hydrology community (Kumar et al., 2021; Sahani, et al., 2019). To address this problem, this

research determines a methodology for selecting, modeling, and evaluating the performance of NBS, and improving the theoretical as well as the modeling aspects of NBS implementation.

The research paper is structured as follows. The case study is presented in Section 2. Section 3 introduces the methodology including the models, NBS scenarios, and performance metrics. The subsequent sections summarize the results of the models, discussions, and conclusions.

## 2 Study case and data collection

The study area, covering 7,000 km$^2$, is the LPD and Lumber rivers watershed, forming part of the lower Pee Dee River Basin located in the Carolinas, USA (Figure 1). These rivers originate in North Carolina's Sandhill area and run south and east to SC's lower Coastal Plain (Howie, 2020). From its headwaters, the Lumber River takes its course of 185 km before merging with the LPD River shortly after entering SC (SCDNR, 2009). Downstream of the confluence, the river continues with the name of LPD River, until it discharges into the Pee Dee River. The LPD River has a length of 140 km from its origin to the watershed outlet. The outlet was defined at the Galivants Ferry bridge a few kilometers downstream of Nichols town and the LPD-Lumber Rivers intersections. At the river's intersection, the Lumber and LPD watersheds have an area of 4,500 km$^2$ and 2,000 km$^2$, respectively. At Galivants Ferry, the annual average discharge is about 78 m$^3$ s$^{-1}$ with a maximum recorded discharge value of 1832 m$^3$ s$^{-1}$, which occurred on the 21 of September 2018 (USGS, 2023).

In the eastern portions of SC and the Coastal Plain, the annual average rainfall ranges from 1143 mm to 1320 mm. Most years, during summer and early fall, SC is affected by tropical storms or hurricanes (DNR, 2021). Hurricanes Matthew and Florence were the latest to create significant flooding in the watershed that was selected for this study.

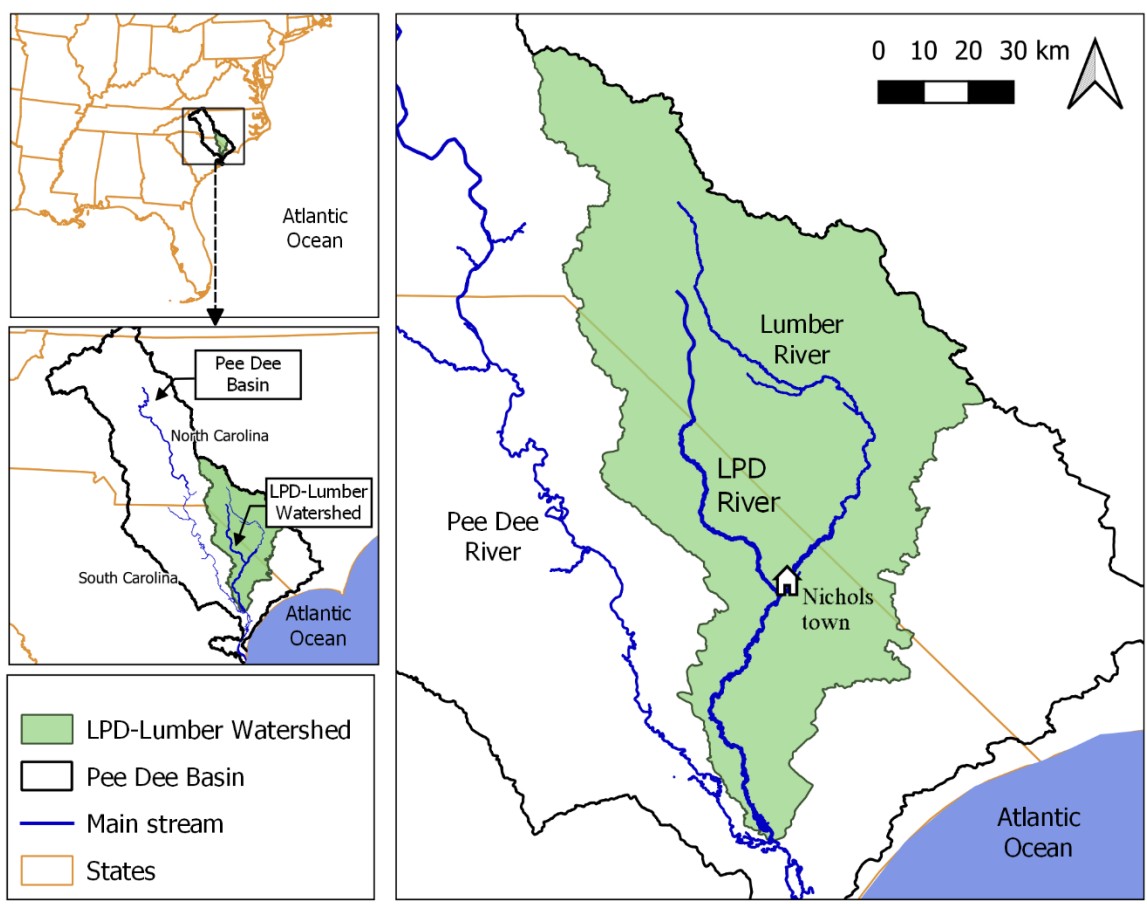

**Figure 1. LPD and Lumber rivers watershed location**

High terrain elevations can be observed at the origins of the Lumber and LPD rivers, reaching up to 225 meters above sea level. However, in the rest of the area, the elevation is low with gentle gradients. The National Elevation Dataset (NED) of the USGS was retrieved and incorporated as terrain data into the models. The terrain elevation data with a resolution of 30 x 30m and 10x 10m were used for hydrological and hydraulic modeling, respectively. All elevation values are in meters and are referenced to the North American Vertical Datum of 1988 (NAVD 88).

The land cover of the watershed is dominated by woody wetlands (32%), followed by agriculture (31%), forests (18%), developed areas (10%), grasslands and shrubs (6%), emergent herbaceous wetlands (2%) and open water (1%). The LULC map is shown in Figure 2. The wetlands and forest ecosystems have experienced high losses since historic dates, with notorious increases in farmland and residential areas. In the past 20 years, the major Land Use Land Cover (LULC) changes were caused by deforestation in woody wetlands and forests. The predominant hydrological soil groups in the watershed are A/D, B/D, and C/D, indicating slow infiltration rates in undrained areas. All these characteristics, gentle slope, dense vegetation, low-relief river, and poorly drained soils, make the area prone to surface ponding with long residence times.

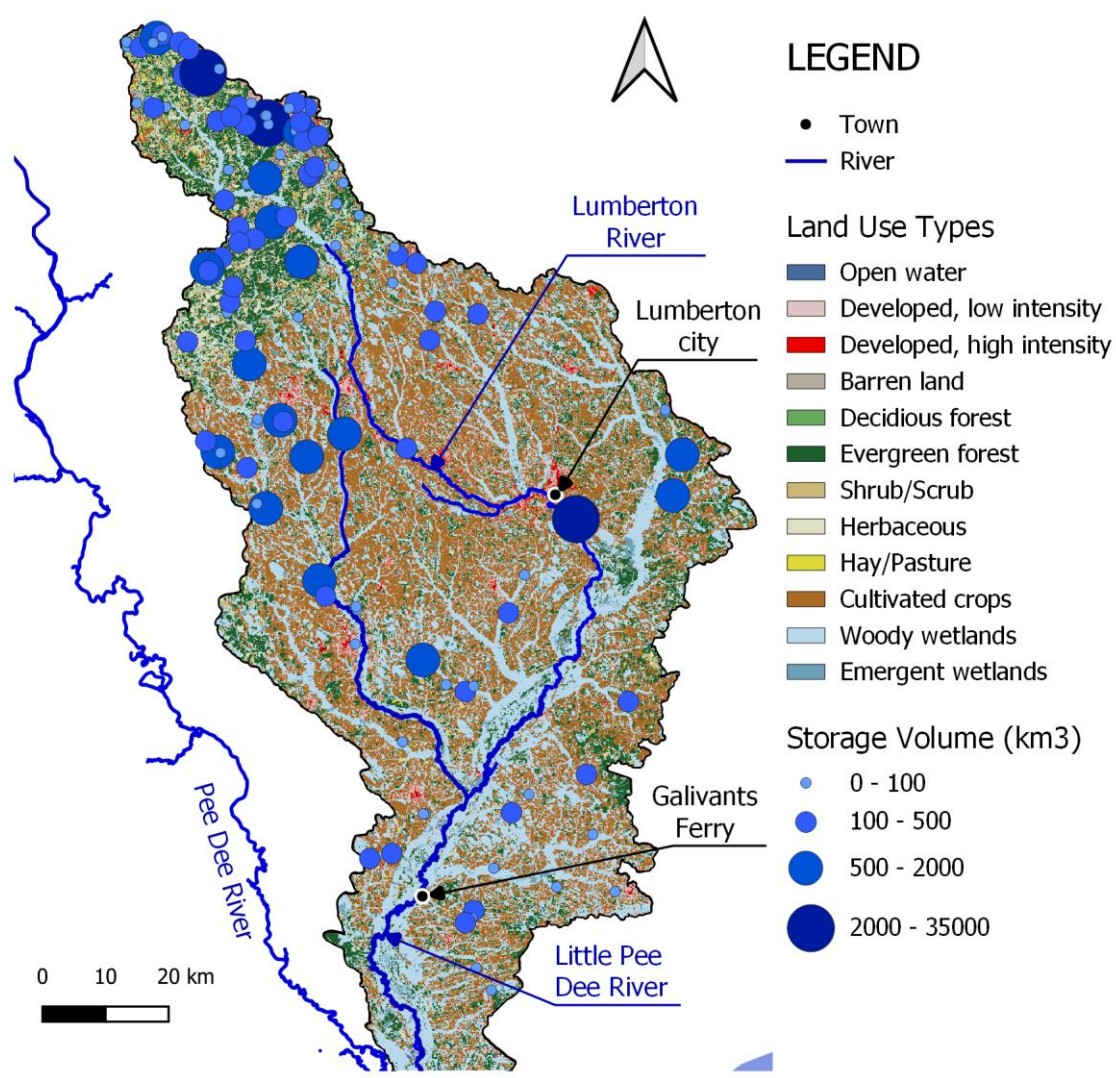

**Figure 2. LULC map and storage areas.**

According to the information retrieved from the National Inventory of Dams (NID), the watershed is covered by storage areas such as reservoirs, lakes, and dams, which can store water volumes during extreme rainfall events (see Fig. 2). However, even in the presence of these topographic terraces, reservoirs, and wetlands, the current storage capacity in the watershed is not enough to avoid destructive flooding events derived from severe storms, like those that occurred during Hurricanes Matthew and Florence.

Local datasets are available from different US institutions, which were used as inputs for the proposed hydrologic and hydraulic models. Table 1 summarizes the data used in this research along with their corresponding sources and resolutions.

**Table 1. List of data used in the study.**

| Data type | Spatial resolution | Time resolution | Source | Usage |
|---|---|---|---|---|
| DEM | $30 \times 30$ m | - | National Elevation Dataset (NED) by USGS | Hydrological and hydraulic simulations |
| | $10 \times 10$ m | | | |
| Gauge rainfall | 12 stations | 24 hours | NOAA | Hydrological simulation |
| | 4 stations | 15 minutes | USGS | |
| Streamflow and water depth gauges | 8 stations | 15 minutes | USGS | Hydrological simulation |
| LULC | 30 m | - | MRLC consortium | Hydrological and hydraulic simulations |
| Soils | 1 km | - | Web Soils Survey by USDA-NRCS | Hydrological simulation |
| Satellite-based flood inundation extent | 250 m | - | Dartmouth Flood Observatory | Hydraulic simulation |

The precipitation data were collected from NOAA and USGS, which have different spatial and temporal resolutions. For Hurricanes Matthew and Florence, the average total precipitation in the watershed according to the NOAA dataset was 287 mm and 388 mm, respectively. The spatial distribution of the rainfall was investigated using Inverse Distance Weighted (IDW) and Kriging methods. The IDW estimates values for unsampled points by the weighted average of observed data at surrounding stations. It relies on the theory that the unknown value of a point is more influenced by closer points than by points further away. Ordinary kriging is a geostatistical interpolation method based on a Gaussian process that considers the spatial variance of the precipitation. The method optimizes the station weights using probability functions and these weights are used to interpolate values for unsampled points across the spatial field (Ly et al, 2011). Interpolation results revealed that cumulative precipitation values during Hurricane Florence were higher in the east of the watershed, whereas for Matthew the maximum volumes appeared in the south.

The streamflow and river stage data are compiled from the National Water Information System of USGS. There are eight streamflow stations with up-to-date data inside the watershed. The newest station was installed next to Nichols town in 2017, after Hurricane Matthew's occurrence. Among all gauges, the Galivants Ferry station, located at the outlet of the watershed, has the most prolonged period of data, with records starting from 1986 to the present. The data shows that the two most extreme flood events of the past 35 years occurred only in the last 5 years. These events were Hurricanes Matthew and Florence, reaching peak discharges of 1,679 $m^3 s^{-1}$ and 1,832 $m^3 s^{-1}$, respectively.

Table 2**Error! Reference source not found.** exhibits information about the USGS stations' metadata (site number, location, contributing area, and peak discharge) during the last major hurricane events. The rainfall and streamflow gauge locations are presented in Fig. 4.

**Table 2. Peak discharge during Hurricanes Florence and Matthew at eight USGS gauging stations (USGS, 2023).**

| USGS site No | USGS site location | Contributing area (km²) | Peak discharge (m³ s⁻¹) | | | |
|---|---|---|---|---|---|---|
| | | | Hurricane Florence | | Hurricane Matthew | |
| 02135000 | Galivants Ferry | 7,226 | 1,832 | 21/09/2018 | 1,679 | 12/10/2016 |
| 02134900 | Nichols | 4,325 | 1,175 | 20/09/2018 | No records | - |
| 02134500 | Boardman | 3,181 | 1,002 | 18/09/2018 | 1,082 | 11/10/2016 |
| 02134170 | Lumberton | 1,834 | 484 | 17/09/2018 | 413 | 10/10/2016 |
| 02133624 | Maxton | 945 | 348 | 17/09/2018 | 192 | 11/10/2016 |
| 02133500 | Hoffman | 474 | 283 | 19/09/2018 | 159 | 09/10/2016 |
| 02134480 | Tar Heel | 593 | 377 | 17/09/2018 | 549 | 10/10/2016 |
| 02132320 | Laurinburg | 216 | 172 | 17/09/2018 | 42 | 09/10/2016 |

The observed flood inundation area was retrieved from remote sensing data. We used Dartmouth Flood Observatory of Colorado University inundation data for Hurricanes Matthew and Florence. The inundation images are from MODIS 250 m and Landsat 8 data were downloaded as raster and converted to flood extent maps. Hurricane Matthew's inundation images were validated by Dartmouth Flood Observatory with Google Satellite images.

## 3 Methodology

To simulate the flooding processes and assess the performance of NBS in the LPD-Lumberton watershed, an integrated hydrological-hydraulic model was developed using the US Army Corps of Engineering software, i.e., Hydrologic Engineering Center Hydrologic Modeling System (HEC-HMS) and River Analysis System (HEC-RAS). Simulations of Hurricanes Matthew and Florence, two of the most damaging flood events in the LPD-Lumber watershed in the past years, were developed and used as case studies for this research. The HEC software have been widely used to assess flood hazards in various US catchments (Bhusal et al., 2022; Knebl et al., 2005; Tang et al., 2020).

Fig. 3 shows the modeling workflow followed in this study to develop an integrated hydrologic-hydraulic modeling system and assess the current situation in the watershed for NBS implementation. The first step was the data collection and pre-processing. We then developed a hydrological model to calibrate hurricane-driven flooding events. The simulated hydrograph was then used as a boundary condition for flood inundation mapping using the HEC-RAS model.

Finally, NBS scenarios were designed and implemented in the models, and steps 2 and 3 were repeated with the required adjustments. Depending on the selected NBS solution, finer DEM might be required. Such is the example of sand traps in a river. The tested solutions in this case study were related to LULC and buffer strips along the river, which were suitable for the DEM resolution of the models used, i.e., $30 \times 30$ m for the HEC-HMS model, and cross sections in the HEC-RAS model based on a $10 \times 10$ m DEM of the channel. New flood maps were obtained and compared with the baseline scenario to evaluate the performance of NBS on flood mitigation at Nichols town.

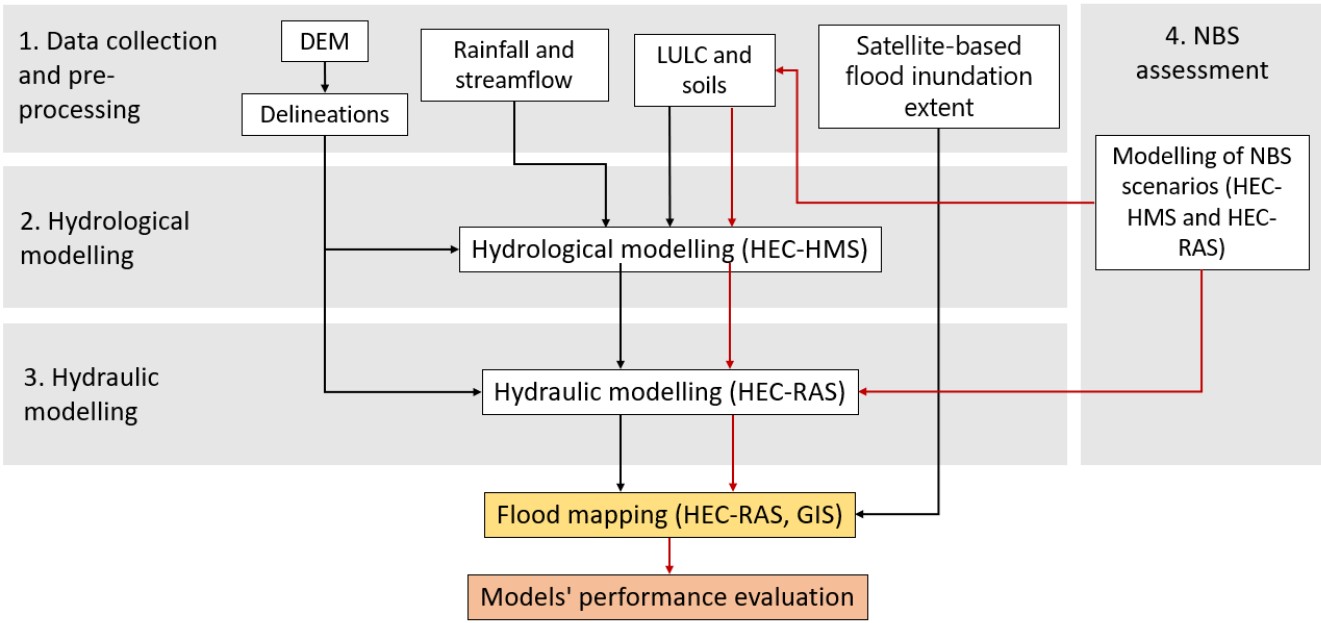

**Figure 3. The workflow of an integrated hydrologic-hydraulic model developed in this research.**

### 3.2 Hydrological modeling

Using HEC-HMS, a hydrological semi-distributed model for the LPD and Lumber rivers watersheds was developed, where each sub-basin is represented as a lumped model. With version HEC-HMS 4.9, the sub-basins and reaches were automatically delineated, using 30 m × 30 m DEM data. In total, the watershed was divided into ten sub-basins with eight of them including a USGS flood gauge at the outlet. The delineated sub-basins, reaches, and flood gauges are shown in Fig. 4.

    The SCS Curve Number (CN) method was chosen to simulate the precipitation loss, Clark Unit Hydrograph (CUH) as the
transformation method, and an Exponential Recession to model the baseflow (US Army Corps of Engineers, 2021).

    The SCS CN method is used for event-based simulation and has been extensively used for US catchments. This approach has been successfully used in 2018 by the North Carolina Emergency Management Agency to simulate the hydrological behavior of Hurricane Matthew in the Lumber River (Emergency Management NC, 2018).

    The model estimates the runoff as the precipitation excess, which is derived from the cumulative precipitation and precipitation
loss. The latter is estimated based on soils, lithology, land cover, and land use data (US Army Corps of Engineers, 2021).

    A weighted average CN number is calculated for each sub-basin based on the hydrologic group and LU types, according to the values in Table 3. As described in the *Study area and data collection* section, the area is covered principally by wetlands, croplands, and forests, and the typical HSG are A/D, B/D, and C/D.

**Table 3. CNs adopted for each LULC and HSG with AMC II (Emergency Management NC, 2018; US Army Corps of Engineers, 2021).**

| LU type | Hydrologic group | | | |
|---|---|---|---|---|
| | **A** | **B** | **C** | **D** |
| Open Water | 100 | 100 | 100 | 100 |
| Developed, Open Space | 49 | 69 | 79 | 84 |
| Developed, Low Intensity | 57 | 72 | 81 | 86 |
| Developed, Medium Intensity | 61 | 75 | 83 | 87 |
| Developed, High Intensity | 77 | 85 | 90 | 92 |
| Barren Land | 77 | 86 | 91 | 94 |
| Deciduous Forest | 36 | 60 | 73 | 79 |
| Evergreen Forest | 36 | 60 | 73 | 79 |
| Mixed Forest | 36 | 60 | 73 | 79 |
| Shrub/Scrub | 35 | 56 | 70 | 77 |
| Herbaceous | 30 | 58 | 71 | 78 |
| Hay/Pasture | 39 | 61 | 74 | 80 |
| Cultivated Crops | 64 | 74 | 81 | 84 |
| Woody Wetlands | 88 | 89 | 90 | 91 |
| Emergent Herbaceous Wetlands | 88 | 89 | 90 | 91 |

The CUH method was adopted to model direct runoff. Clark's model derives a catchment UH by explicitly representing two critical processes in the transformation of excess precipitation to runoff (US Army Corps of Engineers, 2021):

- Translation or movement of the excess from its origin throughout the drainage to the watershed outlet.

- Attenuation or reduction of the magnitude of the discharge as the excess is stored throughout the watershed.

It seems both mechanisms actively control rainfall-runoff processes in our study area. The model requires two input parameters: the time of concentration (Tc), and the storage coefficient I. The Tc was initially estimated using the SCS lag equation developed by the SCS (1972). The SCS recommends that the lag equation (Eq. (1)) be used on basins that may be considered homogeneous in nature and less than 800 hectares in size. Due to these restrictions, the method may be limited for application on most basins, however, it has been widely used and accepted around the world for catchments of varying sizes

(Soulis, 2021; Mishra & Singh, 2003; Knebl et. al., 2004) and it had been used in the study area for a similar application (Emergency Management NC, 2018). We used this approach for an initial estimation of the values. The equation uses the path length of the stream in feet ($L$), the potential maximum retention in inches ($S$), and the average watershed land slope in % ($Y$) to estimate the lag time in hours ($T_{lag}$) (US Army Corps of Engineers, 2021):

$$T_{lag} = \frac{L^{0.8}(S+1)^{0.7}}{1900.Y^{0.5}}$$
(1)

Then, the time of concentration can be derived from $T_{lag} = 0.6 T_c$.

The initial estimation for the storage coefficient I was estimated using the following relationship (US Army Corps of Engineers, 2021):

$$\frac{R}{Tc+R} = 0.6$$
(2)

The Exponential Recession method was used to represent watershed baseflow. It requires the initial flow ($Q_0$), the recession
ratio ($k$), and the ratio to peak as input parameters (US Army Corps of Engineers, 2021).

For channel routing simulation, the Muskingum method was selected. This method uses a simple finite difference discretization of the continuity equation to route an inflow hydrograph. The method aims to capture during the simulation the observed increase and decrease in channel storage during the passing of a flood wave. The required parameters for the method are the travel time ($T$) of the flood wave through any reach and a dimensionless weight $X$ that can range from 0 to 0.5.

Two hurricane types with similar durations and rainfall intensities were considered in this study. Although, the angles of landfall and storm tracks were different for each one of them. In this way, the different spatial and temporal variabilities of hurricanes and how they affect the watershed's response to flooding could be analyzed. The selected hurricanes were Matthew and Florence. Hurricane Matthew had a coast parallel storm track with higher cumulative precipitation volumes in the south of the watershed. In contrast, Hurricane Florence had a meandering storm track with greater precipitation rates on the east side
of the watershed, resulting in extremely large flows in all the tributaries. Hurricane-force winds extended outward up to 110 km from the center during Hurricane Florence (National Hurricane Center, 2018) and up to 75 km during Hurricane Matthew (NASA, 2016). Williams et al. (2020), examined the rainfall volumes during the month of occurrence of the hurricanes and in the previous month. They discovered that rainfall in the watershed one month before Hurricane Matthew was nearly equal to the month of the hurricane, placing wet soil moisture conditions for the hurricane month. On the contrary, before Hurricane
Florence, the watershed received little rainfall, leading to drier soils during the hurricane event.

The two simulated events are separately calibrated. The calibration process attempts to reproduce the peak discharges and total flood volumes at all the flood gauges in the watershed. Calibration was achieved by adjusting the CNs, time of concentration, storage coefficient, baseflow, and channel routing parameters in each sub-basin. The calibration was performed with a time interval of 15 minutes and the calibration periods were:

• Hurricane Matthew model calibration period: 06 October 2016–31 October 2016

• Hurricane Florence model calibration period: 14 September 2018–05 October 2018

### 3.3 Hydraulic modeling

The HEC-RAS version 6.2 was used to develop the hydraulic model. A 1D simulation was performed, which executes surface profile calculations in a gradually varied flow. The program solves water surface profiles from one cross-section to the other
by solving the Energy equation.

The hydraulic model domain was restricted to a small area in the proximity of Nichols town, 20 km of the Lumber River was modeled upstream of the town and 20 km more downstream to include the confluence with the LPD River (see Fig. 4).

To generate the HEC-RAS geometric input data, the Triangular Irregular Network (TIN) was obtained from the 10 × 10m DEM. The main channel, riverbanks, and flow paths were created using georeferenced information from the National
Hydrography Service (NHS) and manually adjusted with the aerial image to follow more precisely the river path. The channel width was visually compared with Google Satellite images. Although there are no measured geometry data of the river and

floodplain cross sections, 3 m × 3 m DEM data is available in some portions of the watershed, and it was used for correcting the main channel sections. The river's cross-sections were generated every 2000 m, and, at Nichols town, the resolution was increased by placing the cross-sections every 1000 m. The upstream boundary conditions for the model are the discharges

simulated with the HEC-HMS in the Lumber River and the LPD River, while the normal depth was used as the downstream boundary condition.

Manning's roughness coefficient was automatically assigned in the cross sections from the Land Use map in HEC-RAS. The software calculates a weighted average of n in each cross-section, taking into consideration the different LULC types. Each LULC type was designated a manning roughness coefficient following Table 4.

(Emergency Management NC, 2018; US Army Corps of Engineers, 2021)

**Table 4.  Manning's roughness coefficients according to LULC type ([1]US Army Corps of Engineers, 2016; [2]NRCS Kansas, 2016).**

| LULC | Normal n value [1] | Allowable range of values [2] |
|---|---|---|
| Developed | 0.12 | 0.03 - 0.2 |
| Evergreen forest | 0.15 | 0.1-0.16 |
| Grassland / herbaceous | 0.035 | 0.025-0.05 |
| Cultivated Crops | 0.05 | 0.025-0.05 |
| Woody wetlands | 0.07 | 0.045-0.15 |
| Emergent herbaceous wetlands | 0.045 | 0.05-0.85 |
| Main channel | 0.035 | |

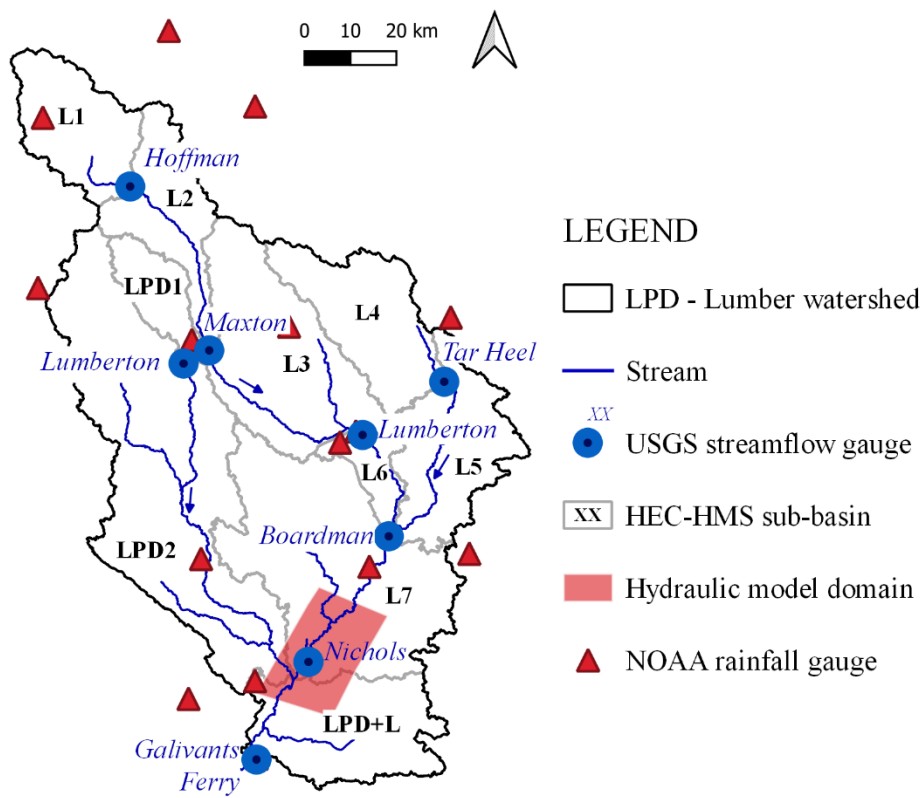

**Figure 4. Sub-basins and hydraulic model domain**

**3.4 Performance metrics**

The validation of the model was assessed by comparing the simulated and observed flood inundation areas from satellite images for Hurricanes Florence and Matthew. The simulated flood inundation maps of both hurricanes were transferred from HEC–RAS to a GIS environment as raster files to process and compare with the observed data. Flood mapping performance is evaluated by using categorical verification statistics, which are usually implemented for estimating the accuracy of flood forecasts (Bhatt et al., 2017; Bhattacharya et al., 2019). The categorical verification statistics measure the correspondence between estimated and observed inundation patterns. In this study, the probability of detection (POD; Eq. (3)), false alarm ratio (FAR; Eq. (4)), and critical success index (CSI; Eq. (5)) were used. The POD indicates what fraction of the observed inundation was correctly simulated. The FAR indicates what fraction of the simulated inundation did not occur. The CSI indicates how well the simulated and observed inundation fit.

$$POD = \frac{hits}{hits+misses} \tag{3}$$

$$FAR = \frac{false\ alarms}{hits+false\ alarms} \tag{4}$$

$$CSI = \frac{hits}{hits+misses+false\ alarms} \tag{5}$$

The observed and simulated inundation polygons were intersected to find areas of hits, misses, and false alarms. These are defined in the following way:

- Hits: When a grid cell in the satellite image shows wet and in the simulated inundation map shows wet.
- False alarm: when a grid cell in the satellite image shows dry and in the simulated inundation map shows wet.
- Misses: When a grid cell in the satellite image shows wet and in the simulated inundation map shows dry.

In addition, This study used a variety of performance metrics for model benchmarking including the Nash-Sutcliffe efficiency (NSE; Eq. (6), Nash and Sutcliffe, 1970), Root Mean Square Error (RMSE; Eq. (7)), correlation coefficient ($R^2$; Eq. (8)) and the flood peak relative error (PE; Eq. (9))

$$NSE = 1 - \frac{\sum_{i-1}^{n}(Pi - Oi)^2}{(Oi - \bar{O})^2} \tag{6}$$

$$RMSE = \sqrt{\frac{\sum_{i=1}^{n}(Oi - Pi)^2}{n}} \tag{7}$$

$$R^2 = \frac{\sum(Oi - \bar{O})(Pi - \bar{P})}{\sqrt{\sum(Oi - \bar{O})^2 + \sum(Pi - \bar{P})^2}} \tag{8}$$

$$PE = \frac{O_{peak} - P_{peak}}{O_{peak}} \tag{9}$$

Where $P$ and $O$ represent simulated and observed stream discharges, respectively.

### 3.5 NBS assessment

The methodology used to assess the implementation of NBS as flood mitigation measures in the watershed consist of the following steps:

i. Selection of NBS measures
ii. Site suitability analysis of selected NBS measures
iii. Development of NBS scenarios
iv. Modeling of scenarios in an integrated hydrologic and hydraulic model

i. Selection of NBS measures

The chosen NBS should be appropriate for implementation in the watershed and feasible to include in the hydrologic and hydraulic models. They should also be feasible to be applied in large and undeveloped areas. Three NBS measures were selected for the study area which have proven to be correctly implemented in HEC-HMS and HEC-RAS models in various cases (Brink et al., 2016; Thomas & Nisbet, 2007). They are offline flood storage ponds, riparian reforestation, and afforestation in croplands.

Flood storage ponds are used to both attenuate the incoming flood peak and to delay the timing of the flood, so the volume is discharged over a longer period. Offline storage is usually located within the floodplain of large rivers with wide floodplains. In these structures, the water is diverted from the river channel, stored, and slowly released back into the river (Ecologic

Institute gemeinnützige, 2019). HEC-RAS v6.2 includes a module intended particularly for incorporating storage areas into the model.

According to the examination of LULC information, there has been a significant decline in forested regions and woody wetlands during the past two decades. Afforestation can help not only reduce runoff volume by enhancing water absorption and interception but also reduce water velocities. The afforestation measures can be very easily implemented in the hydrologic and hydraulic models, compared to other types of natural infrastructure.

ii. Site suitability analysis

A site suitability analysis was conducted for defining the areas where selected NBS can be implemented. The domain for allocation of NBS was considered upstream of Nichols town, in the Lumber River watershed between Nichols and Lumberton towns (Lumber sub-basins 6 and 7), and the Big Swamp Creek (Lumber sub-basins 4 and 5).

According to Mubeen et al. (2021), the most common criteria to determine NBS site suitability are slope, soil type/class, imperviousness, distance from the stream, land use type/zone, urban land use, and road buffer. Based on the results of Mubeen et al., (2021), the following parameters were chosen to determine the suitable areas for the development of storage ponds and riparian reforestation:

- Slope < 5%
- Pervious areas
- Distance from main river < 1000 m or inside floodplain
- Distance from roads > 50 m
- LULC type: outside of forested areas or woody wetlands

The analysis of site suitability was performed in a GIS environment. The slope was derived from the DEM, pervious areas from imperviousness maps, forested areas, and woody wetlands from LULC maps, and stream and road distances were derived from buffers of the georeferenced data. From each criterion, raster maps were developed and then transformed into Boolean maps showing areas where each condition is met. The combination of these maps produces a general suitability map where ponds or riparian vegetation can be allocated. The general suitability map was combined with measuring specific criteria for riparian reforestation, which allowed for the separation of those areas from the storage pond areas (Fig. 5). This criterion is to restore riparian forests to the situation of the year 2001. A separate map for afforestation in croplands was developed; in this case, the suitable areas are all the croplands in the considered sub-basins (Fig. 6).

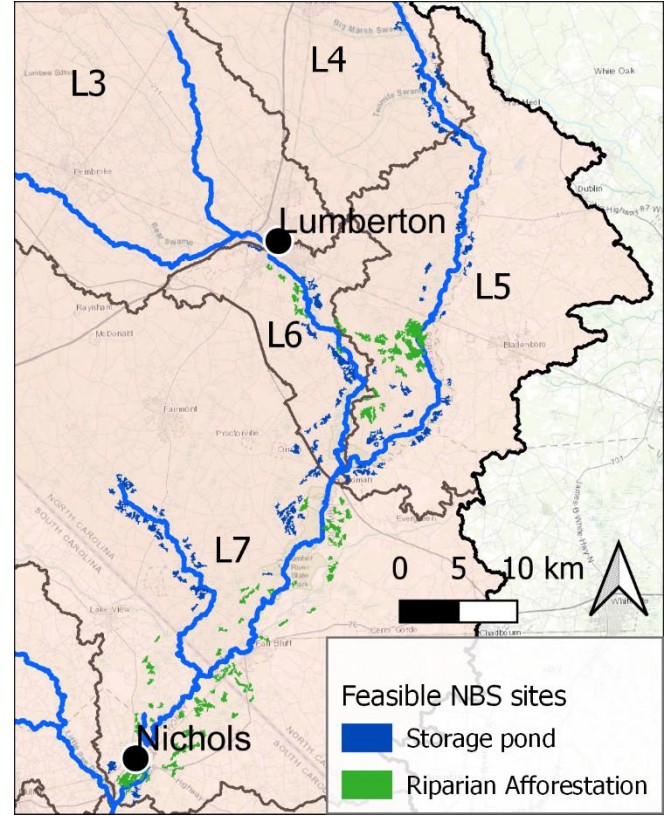

**Figure 5. Feasible sites for storage ponds and riparian reforestation (original LULC layer extracted from MRLC consortium).**

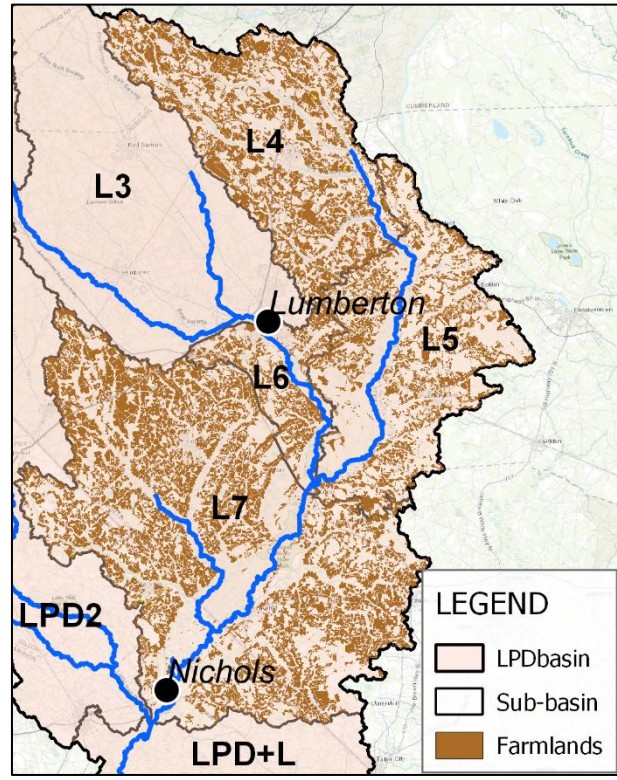

**Figure 6. Feasible sites for afforestation in croplands (original LULC layer extracted from MRLC consortium).**

iii. Development of NBS scenarios

Four NBS scenarios were developed in this research:

- Scenario 1: Offline flood storage pond. This scenario consists of one offline flood storage pond located on the right margin of the Lumber River floodplain near Nichols town. Conceptually, the storage pond would involve constructing a 1 m high berm around the perimeter of the pond area, with inlet and outlet structures from and towards the Lumber River.

- Scenario 2: Riparian reforestation. For the riparian reforestation scenario, forested areas were restored to their
previous state in 2001 inside the floodplain areas. Overall, 23 km$^2$ of the watershed were converted to forests in this scenario, influencing the CN of the watershed.

- Scenario 3: Afforestation in croplands. The conversion of cropland areas to forests also influences the CN of the watershed due to changes in LULC. Three sub-scenarios were developed according to the percentage of cropland area converted to forest:
345        o   Scenario 3a: changing 10% of the cropland area to forest, 96 km$^2$ afforested.
       o   Scenario 3b: changing 20% of the cropland area to forest, 144 km$^2$ afforested.

- o Scenario 3c: changing 50% of the cropland area to forest, 192 km$^2$ afforested.
- Scenario 4: Combination of scenarios 1, 2, and 3. This scenario is the combination of all three proposed NBSs together.

iv. Scenario modeling

The NBS scenarios were tested on the Hurricane Matthew event, given its elevated performance in inundation modeling for the base scenario compared to Hurricane Florence. The same process for flood mapping was followed as in the base scenario but with new adjustments to the hydrologic and hydraulic models. The afforestation scenarios were evaluated with the new CN in HEC-HMS, the new results were used as inputs for the hydraulic model, and new flood maps were obtained. Regarding
the flood pond, the scenario was directly evaluated in HEC-RAS. The Storage Areas, incorporated in the software, create lake-like regions that water can be diverted into or from (US Army Corps of Engineers, 2016). The reduction in discharge and flood inundation over Nichols town were compared with the base scenario to assess the NBS performance.

**4 Results**

This research contributes to the literature on the modelling of NBS measures at a catchment level, which proved useful to
mitigate, to a certain extent, hurricane-driven flooding events. We determined a methodology for selecting, modeling, and evaluating the performance of NBS within a catchment, which can be also extended to other case studies. Another expectation of this research is that it can help decision-makers to define proper Flood Risk Management strategies. The modeling techniques and methodology proposed for the NBS implementation can be of interest to flood/stormwater modelers, and the results can be broadly integrated with the government effort for flood mitigation plans. While severe damages were reported
in many parts of the LPD watershed, a lack of proper studies and research in the area hinders the process of implementing flood reduction measures. This research contributes to identifying at-risk areas that can be potentially stabilized by the NBS measures.

**4.1 Hydrological modeling**

The LPD-Lumber watershed was modeled in HEC-HMS and divided into ten sub-basins. Each sub-basin was considered as a
lumped model where the runoff at the outlet was estimated from the input rainfall and calibrated with a streamflow gauge.
For the SCS CN method, a composite CN value was calculated for each sub-basin, weighting the CNs by the area of different LULC classes in the sub-basin. The LULC classes were assigned a CN based on the tables developed by the SCS and from recommendations of other case studies in the US (US Army Corps of Engineers, 2021). The initial estimations of composite CNs in the sub-basins varied between 57.7 to 80.4.
For the runoff transformation model, the CUH was adopted. The initial estimations for the time of concentration ranged from 26 h to 64 h, while the storage coefficient varied between 36 h to 96 h through the sub-basins.

For the estimation of the baseflow, the Exponential recession method was used. The initial flow is an initial condition estimated as the average flow on the day before the start of the storm. The recession constant $k$ and the ratio to peak were initially estimated at 0.95 and 0.25 respectively, values recommended by the HEC-HMS technical reference manual (US Army Corps of Engineers, 2021). These values were later adjusted during the calibration process.

For the channel routing, the Muskingum method was selected. The travel time of the flood wave through the reach $K$ was estimated as the interval between similar points on the inflow and outflow hydrographs of the sub-basin. The dimensionless weight $X$ is initially estimated at 0.5 as recommended by the HEC-HMS technical reference manual (2021), and its value was later adjusted in the calibration.

The calibration was performed using an automatic process based on the Simplex search algorithm, minimizing the peak weighted RMSE at all streamflow stations. The calibration process consisted of, firstly, defining model parameter constraints and secondly, automatically calibrating each sub-basin progressively from upstream to downstream. The parameter constraints were selected from other case studies, suggestions, and information about the watershed. It is desired that the estimated parameters do not vary far from the estimated values (only around 20%), but individual exceptions were considered to accommodate for the watershed's special characteristics. As indicated before, the watershed has a considerable effect on ponding and water detention due to its topography, vegetation, and soil types. This significantly influences the water volumes that reach the stream and its travel times. To accommodate for this effect, large values were permitted for the storage coefficient and time of concentration, while low values were allowed for CN. The ranges for the recession constant, ratio to peak, and attenuation factor were obtained from recommendations in the HEC-HMS manual (US Army Corps of Engineers, 2021). The calibration procedure was performed by automatically calibrating the parameters of each sub-basin. If the sub-basin had a flood gauge at the outlet, the parameters were calibrated individually. If not, more than one sub-basin was calibrated simultaneously. Calibration was conducted from upstream to downstream in a stepwise manner.

Figures 7, 8, and 9 show the simulated hydrographs for Hurricanes Matthew and Florence at the eight flood gauging stations. In general, the trend of the observed hydrographs was calibrated well in both hurricane models. The obtained calibration accuracies are good according to the performance metrics results shown in Table 5.

**Table 5. Accuracy ranges of hydrological simulations at all flood gauging stations.**

| Variable | Hurricane Matthew | Hurricane Florence |
|---|---|---|
| RMSE/avg | 11%-38% | 13%-64% |
| NSE | 0.89-0.99 | 0.89-0.99 |
| R2 | 0.94 – 0.99 | 0.90 – 0.98 |
| Flood peak error | 2.2% - 7.4% | 7.2% - 11-3% |

The calibrated parameters show specific trends that permit a comparison of both hurricanes' behaviors. Matthew´s model calibration required higher values of CN than Florence´s model, possibly indicating wetter antecedent moisture conditions

(AMC) during the former event. This observation matches other studies regarding these storms in the Lumber River
(Emergency Management NC, 2018; Doll et al., 2020) and agrees with the findings of Williams et al. (2020) highlighting the large rainfall volumes of the month before Hurricane Matthew. Additionally, it was observed that most of the calibrated CN values are in the estimated range between dry and normal AMC in both hurricane simulations. This effect can be attributed to dry AMC in the watershed soils; however, this contradicts the previous findings of a wet month previous to Hurricane Matthew. Another explanation can be attributed to the water detention and ponding effects which are expected to decrease the total runoff
volumes of the watershed. Also, water detention and ponding effects can influence the calibrated storage coefficients and times of concentration, which resulted generally higher than those initially estimated.

The hydrograph at the *Lumberton* flood gauge showed a bimodal behavior with 2 evident flood peaks, one occurring on the day of the peak rainfall and the second occurring between 3 to 4 days later (see Fig. 7). It was assumed that the first peak corresponds to the response of the sub-basin to flooding, while the second peak was generated by the delayed flow originating
from upstream portions. The travel time values in the upper watershed were much larger than expected, with travel times from *Hoffman* to *Maxton*, and from *Maxton* to *Lumberton* gauges ranging between 2 to 3 days. The peaks at *Hoffman, Maxton,* and *Lumberton* were around 40% lower during Hurricane Matthew compared to those of Hurricane Florence. This observation matches the results from the rainfall interpolations that show less concentration of rainfall volumes in the north of the watershed during Hurricane Matthew.

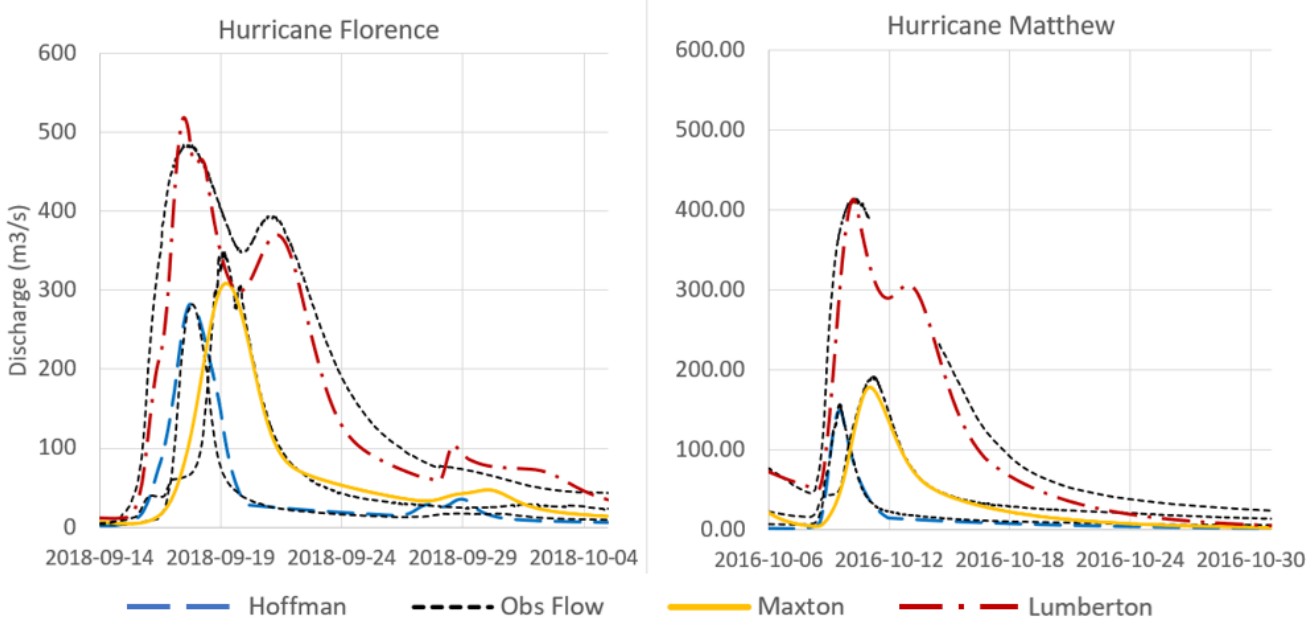


**Figure 7. Observed and simulated flood hydrographs at the Hoffman, Maxton, and Lumberton stream gauges.**

During Hurricane Florence, the first and the highest peak at Lumberton was generated by the *Lumber 3* sub-basin runoff (see Fig. 4). This highest peak, together with the input from the Big Swamp creek (*Tar Heel* hydrograph), created the flood peak

observed at the *Boardman* (Fig. 8). The second and the smaller peak observed at the *Lumberton* gauge does not contribute to the main flooding peak at any of the downstream gauges. However, it caused a slower recession of the hydrograph in the downstream portion of the watershed (Fig. 9), thereby increasing the flooding lead time in that portion.

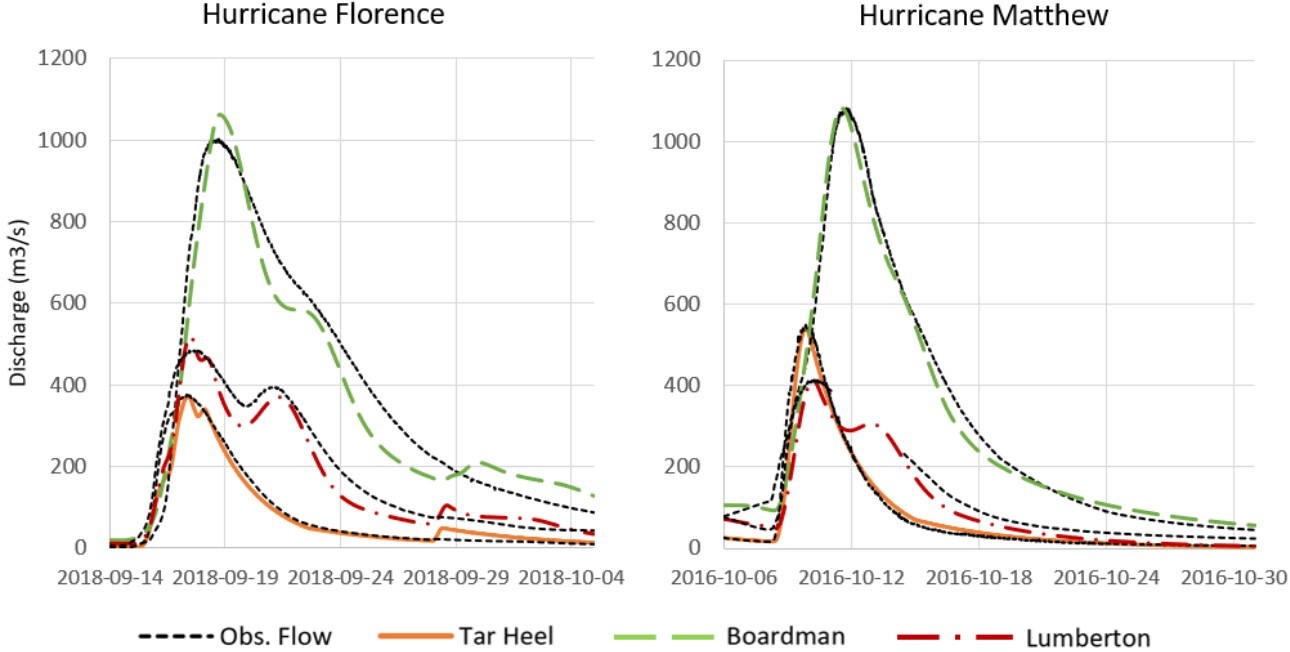

**Figure 8. Observed and simulated flood hydrographs at the Tar Heel, Lumberton, and Boardman stream gauges.**

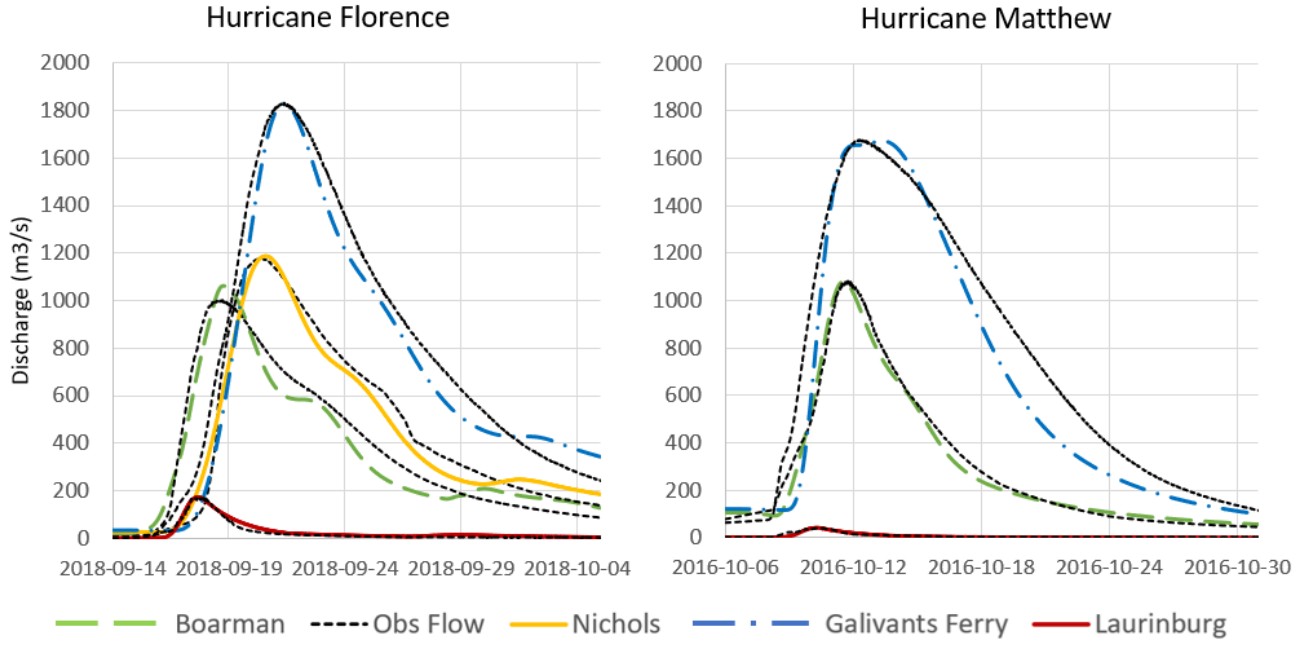

Figure 9. Observed and simulated flood hydrographs at the Laurinburg, Boardman, Nichols, and Galivants Ferry.

This simulation shows that the simultaneity of peak flow occurrence from the LPD and Lumber rivers during Hurricane Florence is one of the reasons for the high flood peak rate at the river's intersection as reported elsewhere (Muller, 2020). During Hurricane Matthew, the LPD River peak arrived 2.5 days earlier than the peak from the Lumber River, while for Hurricane Florence they occurred within a period of just a few hours. The peak flows and time to peak are presented in Table 6.

Table 6. Simulated flood peaks at the LPD and Lumber intersection.

| Simulation | River | Peak flow ($m^3 s^{-1}$) | Peak time |
|---|---|---|---|
| Hurricane Matthew | Lumber | 1,248 | 12 October 2016 23:30 |
| | LPD | 566 | 10 October 2016 14:45 |
| | Lumber + LPD | 1,678 | 12 October 2016 10:45 |
| Hurricane Florence | Lumber | 1,187 | 20 September 2018 15:15 |
| | LPD | 689 | 20 September 2018 2:30 |
| | Lumber + LPD | 1,859 | 20 September 2018 11:00 |

## 4.2 Hydraulic modeling

The hydraulic model requires separate roughness coefficients for floodplains and the main channel. In the floodplains, Manning's roughness coefficient was automatically assigned in the cross sections using the LULC map following Table 7. For the main channel, a Manning coefficient of 0.35 was selected.

**Table 7. Manning's roughness coefficients by LULC type ([1]US Army Corps of Engineers, 2016; [2]NRCS Kansas, 2016).**

| LULC | Normal n value [1] | Allowable range of values [2] |
|---|---|---|
| Developed | 0.12 | 0.03 – 0.2 |
| Evergreen forest | 0.15 | 0.1-0.16 |
| Grassland / herbaceous | 0.035 | 0.025-0.05 |
| Cultivated Crops | 0.05 | 0.025-0.05 |
| Woody wetlands | 0.07 | 0.045-0.15 |
| Emergent herbaceous wetlands | 0.045 | 0.05-0.85 |
| Main channel | 0.035 | |

The calibration process attempts to reproduce the observed peak water level at Nichols for Hurricane Florence. This is managed by manually adjusting Manning´s roughness coefficient of the woody wetlands, which is the land use type with the largest area in the model domain and for which the results show the most significant sensitivity. The water elevation observed at Nichols gauge during Hurricane Florence was 17.0 m above NAVD 88. The calibrated Manning's roughness coefficient of woody wetlands resulted in a value of 0.09, which is within the allowable range of values according to Table 7.

The flood Inundation maps of both hurricanes are shown in Fig. 10. Water depths reached up to 4.5 m and velocities to 1.7 m s$^{-1}$ during Hurricane Florence and up to 4.4 m and 1.6 m s$^{-1}$ during Hurricane Matthew. During both events, Nichols's town was severely inundated causing severe damage to properties, businesses, and residential homes.

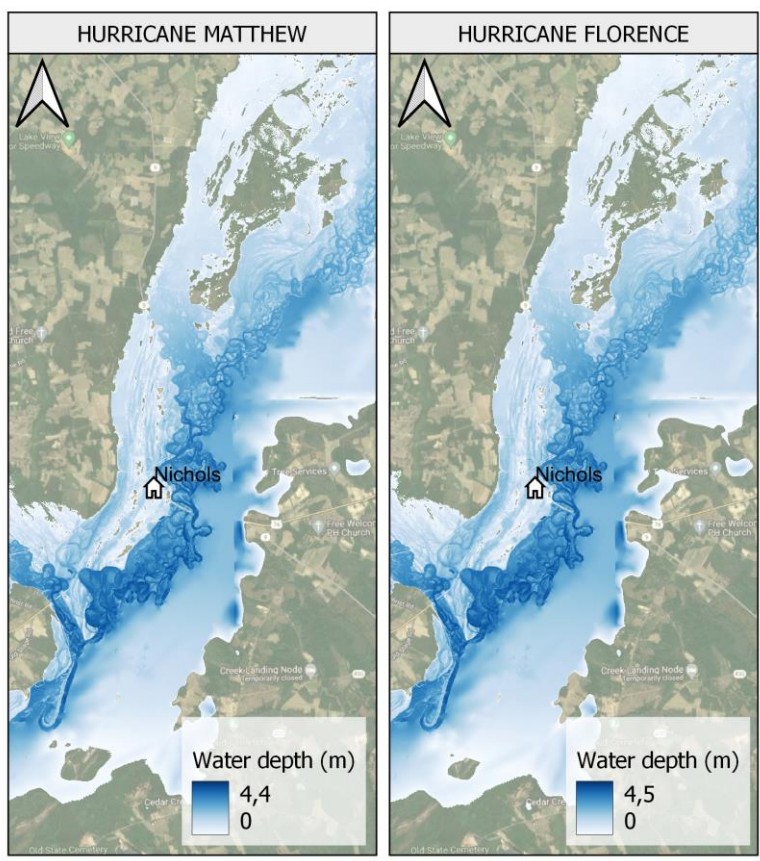

450

**Figure 10. Simulated flood depth during Hurricanes Matthew and Florence in Nichols town.**

## 4.3 Validation of flood inundation

Figure 11 presents the calculated false, correct, and missed alarms that are compared with the observed inundations. These areas are used to calculate the categorical verification statistics which resulted in, for Hurricane Matthew, POD = 79%, FAR = 20%, and CSI = 67%, while for Florence they were POD = 83%, FAR = 51%, and CSI = 45%. The higher the values of CSI and POD are, the more accurate the model is, and the opposite is true for FAR. It can be inferred that the simulated inundation extents showed a good match with the observed satellite images for Hurricane Matthew, and less good for Florence. The PODs are high for both events, while CSI is only high for Matthew. Furthermore, the FAR value is low for Hurricane Matthew, but for Florence is more than 50%.

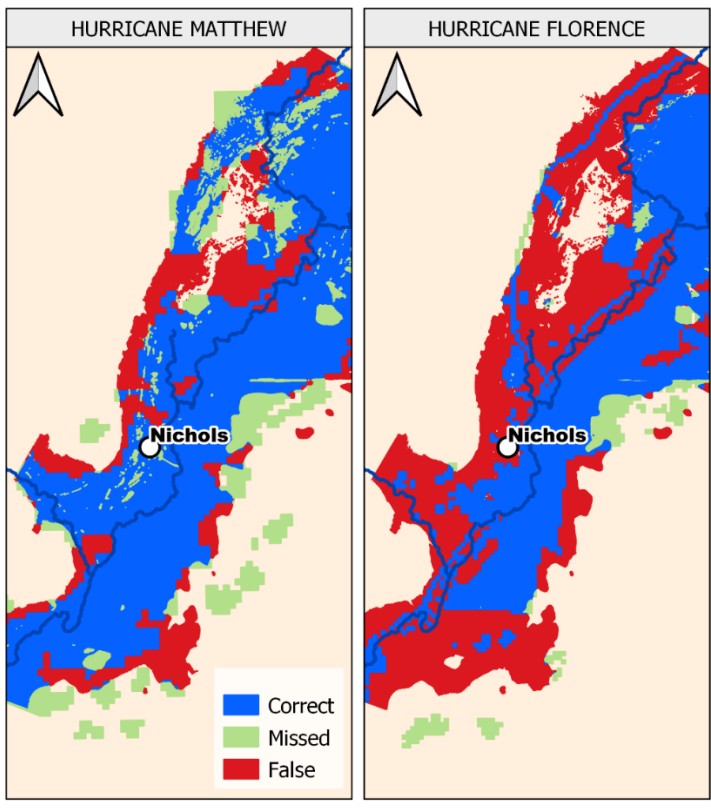

**Figure 11. Correct, missed, and false alarm areas in Nichols town.**

There are several reasons why Hurricane Florence's inundation mapping was less accurate. It is possible that the inaccuracy was produced by errors in the observed satellite image, caused by cloud obstructions during the hurricane event. After analyzing the satellite images provided by the Flood Observatory for the Hurricane Florence event, discordances were found with other observed and measured data. A comparison of both hurricanes' flood inundation maps was conducted to understand the similarities and differences (see Fig. 12). While comparing the satellite observations of Hurricanes Florence and Matthew, it was found that the flooding extent for Matthew's event around Nichols town is much bigger than the one during Florence and that the satellite-observed flood extent of Hurricane Florence does not cover Nichols town, contradicting other more reliable sources of information.

It is also possible that the inaccuracies are caused by the model set-up which could not capture all features of the terrain, such as the existence of structures built along the river, after Hurricane Matthew.

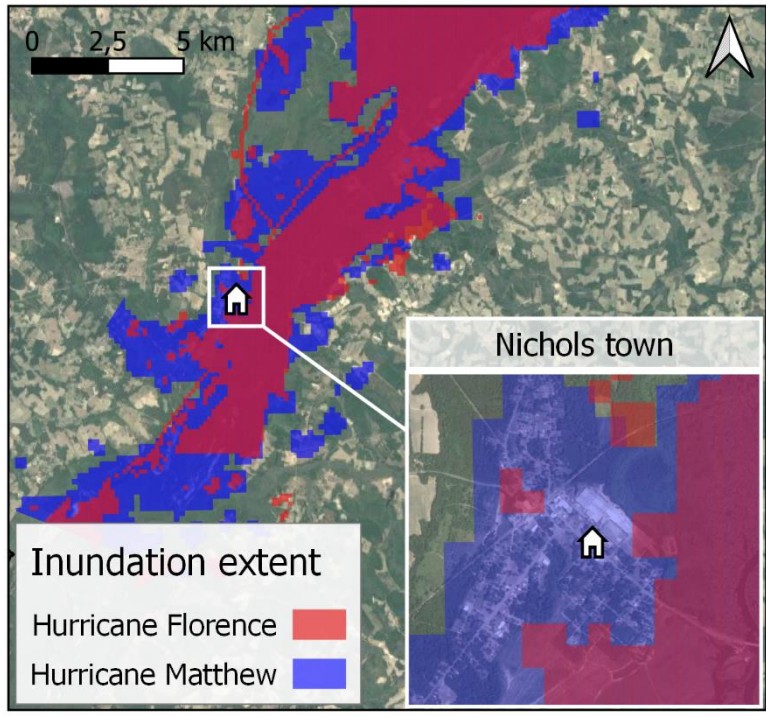

**Figure 12. Observed Flood Extent for Hurricanes Matthew and Florence (layers from Dartmouth Flood Observatory, 2016 & Dartmouth Flood Observatory, 2018).**

### 4.4 NBS performance

The performance of NBS scenarios is evaluated based on their ability to reduce the inundation area at Nichols town for Hurricane Matthew. The total built-in area of the town was estimated at 122 ha. For the base scenario, the model simulation shows that 56% of the town's area was inundated during Hurricane Matthew. With the NBS scenarios, the inundated area was reduced to a range of 55% to 38%. The results for NBS scenarios and the maximum water depth at the USGS gauge are shown in Table 8. The effects of flood storage ponds as well as riparian reforestation are almost neglectable, with an insignificant reduction in flood peak rate and inundation area.

When the afforestation was implemented in a large domain in the model, such as those of scenario 3, the effects start to be visible. These measures showed reductions ranging between 8% and 18% in the inundated urban area. However, still, 38% of the town is at risk of flooding even when afforestation was implemented largely in the model. The reason for this could be that the whole area in the downstream catchments is quite flat, not allowing for great improvements unless great amounts of water are captured upstream. Furthermore, an optimization of the location of these forested areas could improve the results. NBS work better when applied upstream in the catchment. Nevertheless, this study focusses on the effects in downstream areas because Nichol´s town is located there.

Scenario 4 gives comparable results to those in Scenario 3, given that the first two scenarios have an extraordinarily insignificant effect on the results.

With regard to the reduction of flood peak depths, the NBS measure of reforestation on floodplains (Scenario 2) showed an insignificant impact. However, the alternatives in Scenario 3 performed higher peak reductions, ranging between 2.2% and 13%.

**Table 8. Flood reduction performance of the scenarios.**

| Scenario | Area inundated in Nichols (Ha) | % of Nichols area | flood peak depth at Nichols gauge (m) |
|---|---|---|---|
| Base scenario | 68.4 | 56% | 3.15 |
| Scenario 1: Pond | 67.6 | 55% | 3.14 |
| Scenario 2: Riparian reforestation | 66.2 | 54% | 3.12 |
| Scenario 3a: Afforestation, 10% | 59.1 | 48% | 2.96 |
| Scenario 3b: Afforestation, 20% | 56.3 | 46% | 2.94 |
| Scenario 3c: Afforestation, 50% | 46.7 | 38% | 2.84 |
| Scenario 4c: Sc. 1 + Sc. 2 + Sc. 3 | 44.3 | 36% | 2.82 |

## 5 Conclusions and future works

This study developed an integrated hydrologic-hydraulic modeling system for hurricane-driven flood simulation and discussed how NBS measures can be implemented in the proposed modeling system to mitigate flood peak rates and inundation areas. The present study can serve as an example of application and methodology to areas with similar physical characteristics in the world. The catchment characteristics that are important when replicating the methodology are large storage and long detention times, which are also affected by hurricanes or similar storms. Data availability needs to be similarly rich. The extension of these models to less intense storms, or areas with steeper slopes needs to be further investigated. Furthermore, this research adds to the scientific literature on modelling NBS measures at a catchment level, which proved useful to mitigate hurricane-driven flooding events.

Despite the challenges associated with flood modeling in the LPD - Lumber River watershed, the model performed well. However, different complexities made modeling a challenging task. The watershed is a challenging hydrologic system due to its physical characteristics, which make it prone to surface ponding conditions with long residence times. The effects of complex floodplain response, dense vegetation, gentle slope that increases flood ponding/residence time, uncertainty in rainfall and flood variability/distribution as well as reservoir operation can significantly impact the modeling outcomes. Although, the calibration of Hurricanes Florence and Matthew in separate models was performed with good accuracy.

It was discovered that calibrated CN values for both events were lower than those predicted in most sub-basins for normal AMC. A severe dry AMC in the watershed could be the cause of this or, perhaps (since no evidence suggests dry AMC), the Pee Dee area's typical attenuation and detention effects. The flow attenuation effects may also explain the long response times observed in most of the sub-basins. The time between peak occurrence at consecutive flood stations varies across rainfall events, which may be attributed to differences in reservoir operations and initial water content in detention areas.

Water attenuation and detention effects in the upper sub-basins of the Lumber River have a direct impact on increasing the recession periods of the hydrographs in the lower sub-basins. The delayed response from the upper Lumber River created a double peak effect at the Lumberton flood gauge. In both hurricane events, the first and the highest peak rates were generated at the Boardman outlet which largely raised the total flood peak downstream. While the second peak contributed to generating a smooth and slow decrease at the falling limb of the hydrograph, lengthening the time the river takes to get back to its normal flow.

The spatial and temporal patterns of rainfall for both hurricane events play a key role in the generation of flood hydrographs. Higher rainfall volumes concentrated in the south of the watershed during Hurricane Matthew implied that the flood peak at Nichols was mostly due to the Lumber River and less to the LPD. On the contrary, during Hurricane Florence, the whole watershed received higher rainfall amounts, principally in the east. This generated a higher flood peak at the intersection of the LPD and Lumber rivers, caused by simultaneous peaks originating from both rivers.

The simultaneous occurrence of flood peaks from the LPD and Lumber rivers caused the inundation extension to be larger during Hurricane Florence than during Hurricane Matthew. According to the simulated events, Hurricane Matthew inundated 58 % of Nichols town's residential area, whereas Hurricane Florence inundated 91%. In the model domain, simulated water depths and velocities reached 4.5 m and 1.7 m s$^{-1}$ for Hurricane Florence, and 4.4 m and 1.6 m s$^{-1}$ for Hurricane Matthew.

The performance of flood inundation mapping revealed high accuracy for Hurricane Matthew whilst showing less accurate results for Hurricane Florence. The deficient performance of Hurricane Florence is attributed to the observed data rather than simulation errors. The flood extent observations during Florence were not validated against other satellite images due to cloud coverage, as the images mismatched the archived photos of Florence-driven inundation and neighbor testimony.

The NBS scenarios simulation indicated neglectable flood reduction outcomes for the flood storage pond and riparian reforestation, while afforestation in the cropland scenario had a more visible effect. The storage pond worked correctly throughout the simulation, however, the flow that must be decreased to minimize inundation in Nichols town is overwhelmingly high to manage using a single flood storage pond. The NBS site suitability analysis revealed more potential locations outside of Nichols town where ponds can be designed and implemented, allowing for a higher number of storages to be considered.

The afforestation in croplands scenarios revealed the best flood reduction results of any intervention tested. This is because it operates at a much higher scale, which is required in this situation to observe an effect on flood reduction. Evidently, the more cropland area converted to forests, the more reduction in flood peak rate and volume were observed. Although, these findings should be accompanied by an evaluation of the cropland surface area that can be realistically transformed into forests.

The selected loss method (SCS CN) in the HEC-HMS model did not allow for the consideration of the sub-basins' initial conditions in terms of soil moisture (unless CNs are adjusted for each event) and water table elevation. In addition, the HEC-HMS model is unable to account for reservoirs or other detention areas in a direct manner, and the initial water contents of these areas, which could potentially influence simulation results, are uncertain. To account for these uncertainties, water detention in the sub-basins is manually formulated into an equation in the calibrated parameters of CN, time of concentration, and storage coefficients. In this region, dense vegetation and extensive forested/non-forested wetlands may severely impact runoff generation. These complexities along with poor natural drainage created an excessive soil water condition that is difficult for the model to understand and capture their dynamics and interaction. Dense vegetation and forested/non-forested wetlands affect the variable source area involved in the generation of saturation overland flow and this mechanism can cause the rainfall-runoff module to behave chaotically during flood simulation.

Another difficulty that concerned us was the lack of sub-daily rainfall data at some sub-basins which made the modeling and understanding of the runoff generation mechanism difficult. In this case, it is important to model the watershed for other significant flooding events when the rainfall distribution over the watershed is different from Hurricanes Matthew and Florence. In addition, several modeling improvements could be explored in the future. For example, a continuous simulation with a soil moisture accounting model as a loss method could be tested to help improve modeling results.

The large quantity of storage facilities within the catchment likely has a substantial impact on the modelling process. We recognize a limitation in our study regarding the functioning of these reservoirs. Due to the sensitive nature of operating such structures during hazardous conditions, information regarding their operation is not accessible to the public.

We acknowledge that both hydrologic and hydraulic simulations can be made more efficient by coupling these models with Bayesian uncertainty inference such as Bayesian Model Averaging (BMA; Samadi et al., 2020) and/or Markov Chain Monte-Carlo (MCMC) optimization methods (Duane et al., 1987). The analyses presented herein are intended to provide a basis for NBS implementation and assessment in both hydrologic and hydraulic settings. However, subsequent in-depth studies are needed to examine the impacts of individual and combined NBS measures on flood peak and volume reduction. Additionally, checking the emergent behavior of the hydrograph (in more detail) over time using carefully designed in-filed NBS implementation is useful to expose key runoff generation mechanisms in at-risk watersheds. Acknowledging a growing enthusiasm for NBS modeling studies in the hydrology community, we expect progress on multiple fronts: a better module to design NBS structures in the model, better accuracy metrics for quantifying NBS performance, and better error estimation of NBS implementation. As always, we invite dialogue with hydrology communities interested in this and another related modeling for flood risk management.

**Author contributions**

BG, IP, VS, and BB conceptualized and designed the study. BG did the methodology and draft paper write-up. BG and IP did the first figure preparations. Improvements, final reviews, and edits were done by IP, VS, and BB. All authors undertook the analysis.

## Competing interests

The authors declare that they have no conflict of interest.

## Acknowledgments and data

Betina Guido was supported by Grant number 2018-1514lpd 001-001-EMJMD (Flood Risk Management) of the European Commission and Hydroinformatics Research Fund of IHE Delft. Vidya Samadi acknowledges support from the National Science Foundation under grant number CMMI 2125283.

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
