# Peer review of "An integrated modeling approach to evaluate the impacts of naturebased solutions of flood mitigation across a small watershed in the southeast United States"

_Natural Hazards and Earth System Sciences, 2022_

## Referee Comment (RC1)

**An integrated modeling approach to evaluate the impacts of nature-based solutions of flood mitigation across a small watershed in the southeast United States**

Betina I. Guido[1], Ioana Popescu[1], Vidya Samadi[2], Biswa Bhattacharya[1]

5   [1]Department of Hydroinformatics and Socio-Technical Innovation, IHE Delft Institute for Water Education, Delft, 2611 AX, the Netherlands.
[2]Department of Agricultural Sciences, Clemson University, Clemson, SC 29634, United States.

*Correspondence to*: Betina Guido (betinaguido@gmail.com)

The study has well demonstrated the advances of the flood decision-making system. The applied approach is beneficial for adopting non-structural flood mitigation measures. The work is carried out under hydrologic and hydrodynamic modeling and provides the solution for adopting flood mitigation planning in a small watershed in the US. The paper is well written; however, there are some concerns where the author(s) has to explain or justify more for the reader(s) understanding.

The specific comments to improve the work:

Major:
1. It is observed that the study reach is well established hydrologic network. It has enough observed river gauge stations to have input the upstream boundary condition at Hydraulic modeling. Then why author has performed the HEC-HMS hydrologic model to simulate and estimate the flood hydrograph at required location. Furthermore, HEC-HMS is usually performed when the watershed is ungauged and not having enough river-gauges which can utilize to fix the boundary condition of HEC-RAS.
2. What is the probability of detection (POD; Eq. (3)), false alarm ratio (FAR; Eq. (4)), and 235 critical success index (CSI; Eq. (5))? Explain more in detail in part of the Methodology.
3. Lines 325 to 340 are part of the Methodology rather than part of the results.
4. The CN value rest some extent on Soil characteristics. Therefore, the part of the Methodology needs to include a description of the soil map and relational of CN value.
5. How author's has considered the "n" value before and after the scenario.
6. What is court number, how author's has calculated the value for HEC-RAS unsteady flow simulation?
7. What is the 2D grid cell, and how has the author calculated the 2D cells? What is the simulation time?
8. How NBA site is selected, and how this layer added in the model for modeling?

9. The author has mentioned the number of storage structures. The storage structure map is missing? Second, how these structures operated during flooding also significantly influences modeling work.

Minor:
1. Fig. 9 Scale bar and Fig. 4 Scale bar are missing.
2. LU/LC map is missing.

---

## Author Response (AR1)

[revised manuscript text omitted]

**AUTHOR'S REPLY**

**Dear Editor and Reviewers,**

We are pleased that you offered us an invitation to improve our work for your journal; we are delighted to submit this revised version. Your choice of qualified reviewers for our manuscript provided us with insightful comments that contributed to our revision. We closely followed the reviewer's suggestions in this revision. Their comments helped us to improve the clarity of the manuscript and its implications applied to NBS application in the context of hurricane-driven floods.

We believe that the manuscript has been greatly improved and hope it has reached your journal's standard. The methodology presented herein is a real-world application addressing NBS application for persistent hurricane-driven flood issues in the southeast US. We thus believe this paper makes a very good contribution to the readers of this journal given the novelty and the findings of the research. In addition, this research can shed light on how NBS could be potentially formulated in the hydrologic-hydraulic models.

As we revised the paper, we worked hard to provide a more balanced presentation of our arguments and results. It is our intention, after all, to promote a discussion that will help engineering hydrology as a whole to continue its remarkable growth and importance in society. We hope that our revision, based on the input of the reviewers, accomplishes this objective, and addresses your concerns such that you and the reviewers now deem it worthy of publication in your journal.

Once again, we acknowledge the comments very much, which contributed significantly towards improving the quality of our manuscript. Our reply to each reviewer's suggestions can be found below.

**Sincerely,**

**Betina Cruz—the corresponding author**

**REFEREE #1**

*The study has well demonstrated the advances of the flood decision-making system. The applied approach is beneficial for adopting non-structural flood mitigation measures. The work is carried out under hydrologic and hydrodynamic modeling and provides the solution for adopting flood mitigation planning in a small watershed in the US. The paper is well written; however, there are some concerns where the author(s) has to explain or justify more for the reader(s) understanding.*

*The specific comments to improve the work:*

*Major:*

1. *It is observed that the study reach is well established hydrologic network. It has enough observed river gauge stations to have input the upstream boundary condition at Hydraulic modeling. Then why author has performed the HEC-HMS hydrologic model to simulate and estimate the flood hydrograph at the required location? Furthermore, HEC-HMS is usually performed when the watershed is ungauged and not having enough river gauges which can utilize to fix the boundary condition of HEC-RAS.*

   **Response:** You raised a valid point. The goal of this study was to use an integrated hydrologic-hydraulic model to understand the system behavior during flood generation mechanism and study at-risk areas where nature-based solutions could stabilize them. Therefore, we first simulated the flood hydrograph using the HEC-HMS model to understand flood generation mechanisms and study flood hydrograph behaviors over time. The outcome of HEC-HMS was then used as a boundary condition in the HEC-RAS model. We also compared CNs, time of concentration, etc. to understand which parts of the catchment contribute more runoff to the river system. During the calibration of the hydrological model, all the discharge gauges were used and as a result, the hydrologic model is updated with the measured discharge data.

2. *What is the probability of detection (POD; Eq. (3)), false alarm ratio (FAR; Eq. (4)), and 235 critical success index (CSI; Eq. (5))? Explain more in detail in part of the Methodology.*

   **Response:** Excellent question. Indeed the hits, misses, and false alarms are not clearly defined. The manuscript is updated with the following information:

   *Hits*: When a grid cell in the satellite image shows wet and in the simulated inundation map shows wet;

   *False alarm*: when a grid cell in the satellite image shows dry and in the simulated inundation map shows wet;

   *Misses*: When a grid cell in the satellite image shows wet and in the simulated inundation map shows dry.

3. *Lines 325 to 340 are part of the Methodology rather than part of the results.*

   **Response:** Thank you for pointing out this important element of the methodology. The updated version of the manuscript has these lines moved to the Methodology section.

4. *The CN value rest to some extent on Soil characteristics. Therefore, the part of the Methodology needs to include a description of the soil map and relational of CN value.*

   **Response:** This is now included in the paper. Thank you for this direction.

5. *How the author has considered the "n" value before and after the scenario.*

   **Response:** Excellent point. The n before the scenarios is calculated with a weighted average in each cross-section, taking into consideration the different LU/LC types from the LU/LC map. Each LU/LC type is assigned a Manning coefficient value, which is used by HEC-RAS to calculate a weighted average for the cross-section.

   In our case study the Manning coefficient n after the scenario was implemented in the hydrological model only (i.e. HEC-HMS). The reforestation scenarios were only implemented in the hydrological model because the hydraulic model extent was limited to a small part around the affected inundation area. The hydraulic model extent was too small to have implemented the reforestation scenarios on it. This is an important point to improve in future studies and will be included in the Conclusion part of the updated manuscript.

6. *What is court number, how author's has calculated the value for HEC-RAS unsteady flow simulation?*

**Response:** Thank you for this direction. The HEC-RAS program automatically computes the Courant number during a simulation run. HEC-RAS is defined as such that if the Courant number is outside the stability values, computation stops, and an error message will be issued. That was not the case for our computations.

7. *What is the 2D grid cell, and how has the author calculated the 2D cells? What is the simulation time?*

**Response:** This is a great question. This study does not report on HEC-RAS 2D flooding. All implemented NBS required 1D HEC-RAS implementation. The 2D part represented in Figure 10 is not computed under 2D conditions. We implemented RAS Mapper, through which the mapping of the 1D results over the terrain is performed, by interpolating the results obtained at cross sections. This is shown over the terrain but is not the result of 2D flow computations, no 2D domains were considered in this analysis.

8. *How NBA site is selected, and how this layer added to the model for modeling?*

**Response:** Thank you for this direction. We made the assumption that NBA refers to NBS. If this is not the case, we kindly ask the reviewer to let us know what the acronym stands for.

The NBS sites are selected depending on the following characteristics: slope, soil type/class, imperviousness, distance from the stream, land use type/zone, urban land use, and road buffer, where:

- Slope < 5%
- Pervious areas
- Distance from main river < 1000 m or inside floodplain
- Distance from roads > 50 m
- LULC type: outside of forested areas or woody wetlands

With these criteria, suitability maps were developed in QGIS. The NBS selection is explained in the new version of the manuscript in an improved manner.

In the case of reforestation alternatives, no layers are added to the models. The CNs are changed in the HEC-HMS model by making a new weighted average of the CN with the new LU/LC types in each sub-basin. The layers of the changed LU/LC areas are managed in GIS, and results are transferred to HEC-HMS. For the implementation of the flood storage pond, the layer was created first in GIS and uploaded as a storage area in HEC-RAS, after which the depth and spillways were added/ adjusted.

9. *The author has mentioned the number of storage structures. The storage structure map is missing? Second, how these structures operated during flooding also significantly influences modeling work.*

**Response:** we agree with your assessment, and we included storage structures in the updated manuscript. We agree with the Reviewer that the operation of these structures significantly influences the response of the catchment. However, due to the sensitivity of operating such structures in hazardous conditions, the information on reservoir operation is not publicly available, unfortunately, we could not include it in this study.

*Minor:*

1. *9 Scale bar and Fig. 4 Scale bar are missing.*

**Response:** Thank you for pointing this out, this is now updated in the manuscript.

*2. LU/LC map is missing.*

**Response:** Good catch. LU/LC map is now included in the manuscript.

We thank the reviewer for his/her feedback. We found your comments constructive and kind.

840 **REFEREE #2**

*The paper is relevant to the scope of the journal. The paper is well written and narrated with appropriate justification, however, some flaws are observed. It will be included in the revised version to improve the quality of the paper.*

*The specific comments/suggestions for improvement.*

*1. Usually the data collection is part of the study area, it is preferable to include section 3.1 to be a part of section 2.*
845 *It will be included with a new title as study area and data collection.*

**Response:** Thank you for pointing out this important element. The updated version of the manuscript will have section

3.1 incorporated as suggested in section 2.

*2. Number of rainfall data are utilized, however, its location is missing, therefore, it is suggested to include the rainfall location maps in section 2. It would be more meaningful if author include the river gauge and rain gauge in*
850 *the same map for hydrologic parameters understanding.*

**Response:** we agree with your assessment. The updated version of the manuscript will include the rain gauges in Figure 4, together with the streamflow gauges.

*3. Author(s) has utilized IDW and Kriging methods for finding average rainfall or rainfall distribution, however, the important method description in the study is missing.*

855 **Response:** Great direction. This description is now included in the manuscript.

*4. Table 2 there is no major difference of the collection of data in two different storms, however, there is a significant difference of collection of data 377 and 42 by the station Tar Heel in two different storms. What is the specific reason?*

860 **Response:** Thank you for highlighting this, there was an error in this value. The values of rainfall for Tar Heel and Laurinburg were exchanged in the column of Hurricane Matthew. With this change, the differences are smaller. However, there is still a difference in the Laurinburg station, with 172 for Florence and 42 for Matthew. But the reason for this is most surely the rainfall records. In this area the rainfall values for both hurricanes are also very different: 492mm vs 163mm of accumulated rain during Florence and Matthew respectively. The values on the table were updated and all the other values in the table were checked again and no other errors were found.

865 *5. Observed flood inundation is retrieved using remote sensing techniques, however, the satellite information to utilize the flood footprint is missing.*

**Response:** This information is now added in the updated version of the manuscript both in the main text and in references.  Remote sensing data is freely available at: https://floodobservatory.colorado.edu/Events/4676/2018USA4676.html

870

Thank you for pointing out this missing detail.

6.  *Author(s) has validated 2D flood with flood observatory? Then why it is not included as part of the methodology?*

**Response:** This is now included in the performance metrics part of the methodology, with the following text:

875  "The validation of the model was assessed by comparing the simulated and observed flood inundation areas from satellite images for Hurricanes Florence and Matthew."

7.  *Author(s) has applied DEM of 30 m x 30 m in size, however, the source of DEM and detailed description of DEM is missing.*

**Response:** GREAT direction. This info is now added to the bibliography. The following description of the DEM is also added to the manuscript:

880  "The National Elevation Dataset (NED) of the USGS was retrieved and incorporated as terrain data into the models. The terrain elevation data with a resolution of 30 x 30m and 10x 10m were used for hydrological and hydraulic modeling, respectively. All elevation values are in meters and are referenced to the North American Vertical Datum of 1988 (NAVD 88).

885  High terrain elevations can be observed at the origins of the Lumber and LPD rivers, reaching up to 225 meters above sea level. In the rest of the area, the elevation is generally low, and the gradients are gentler, reaching almost sea level in the downstream portions."

8.  *What is CUH method and how it estimates the runoff? It needs to include in the methodology part.*

**Response:** You raised a valid point.

Clark's model derives a catchment UH by explicitly representing two critical processes in the transformation of excess precipitation to runoff:

890

- Translation or movement of the excess from its origin throughout the drainage to the watershed outlet.
- Attenuation or reduction of the magnitude of the discharge as the excess is stored throughout the watershed.

It seems both mechanisms actively control rainfall-runoff processes in our study area. This is now updated in the manuscript.

895  9.  *What mean of georeferenced information and what mean of manually adjusted in GIS environment?*

**Response:** Thank you for your question. The flow paths georeferenced data was obtained from the National Hydrography Service. These layers were manually adjusted with the aerial image to follow more precisely the river path. This info is now included in the updated manuscript.

10. *Author(s) has applied the river cross section at every 2000 m. Interval is too extensive and not more help to simulate the flow in 1D. It should verify once again.*

900

**Response:** Excellent point. The river cross-sections are at a 2000m distance one from another, as the river does not change much in this interval. As the model is built, the HEC-RAS model computes the Courant number and if there is a need to have a more refined space step it gives an error message. Moreover, the interpolated Xsections for the space step of computation were selected to be 500m.

905     11. *Major flaw in the paper is modeling description. Author(s) has utilized the hydrodynamic model for simulation of flow in 1D and 2D, however the basics of modeling and modeling governing equation (i.e. 2D Saint-Venant equation) is missing. Also important modeling parameters which help to understand the modeling simulation process in 2D i.e. Courant Number, average cell size, Computataional time step(s) is missing.*

**Response:** Great question. This study does not have a 2D discretization HEC-RAS model, hence no 2D equations are
910     added. All implemented NBS required 1D HEC-RAS implementation. The 2D part represented in Figure 10 is not computed under 2D conditions. We implemented RAS Mapper, through which the mapping of the 1D results over the terrain is performed, by interpolating the results obtained at cross sections. This is shown over the terrain but is not the result of 2D flow computations, no 2D domain was considered in this analysis.

915     Some References: 1) Pathan, A.; Kantamaneni, K.; Agnihotri, P.; Patel, D.; Said, S.; Singh, S.K. (2022), " Integrated Flood Risk Management Approach Using Mesh Grid Stability and Hydrodynamic Model." *Sustainability 2022, MDPI* 14, 16401. https://doi.org/10.3390/su142416401; 2) Shah Z, Saraswat, Samal D. and Patel DP (2022), "A Single Interface for Rainfall-Runoff Simulation and Flood Assessment – A Case of New Capability of HEC-RAS for Flood Assessment and Management", *Arabian Journal of Geosciences*, *Springer*. 15, 1526(2022) https://doi.org/10.1007/s12517-022-10721-2. 3)
920     Pathan, A.I., Agnihotri P.G , and Patel, D.P. (2022) " Integrated approach of AHP and TOPSIS (MCDM) techniques and GIS for dam site suitability mapping : a case study of Navsari City, Gujarat, *India. Environmental Earth Science*, Springer. (2022) 81:443. https://doi.org/10.1007/s12665-022-10568-6. 4) Pathan A., Agnihotri PG, Patel DP, and Prieto C. (2022), "Mesh grid stability and its impact on flood inundation through (2D) hydrodynamic HEC-RAS model with special use of Big Data platform—a study on Purna River of Navsari city", *Arabian Journal of Geosciences*, *Springer*, 2022 15:659,
925     https://doi.org/10.1007/s12517-022-09813-w.

**Response:** Thank you for giving the references that were the basis for this review. We read the ones to which we had access. While they are very interesting, we could not fit their presentations with our paper, but we will have them on the list for the next paper on the flooding with HEC-RAS. Greatly appreciated.

930     We thank the reviewer for his/her feedback. We found your comments constructive and kind.